# Systematic Correlation Matrix Evaluation (SCoMaE) - A bottom-up, science-led approach to identify Indicators

Nadine Mengis[1,2], David P. Keller[2], and Andreas Oschlies[2,3]

[1]Concordia University, Montreal, QC, Canada
[2]Helmholtz Center for Ocean Research Kiel (GEOMAR), Düsternbrooker Weg 20, 24105 Kiel, Germany
[3]Kiel University, D-24098 Kiel, Germany

*Correspondence to:* N. Mengis, Concordia University, Montreal, QC, Canada, *nadine.mengis@concordia.ca*

**Abstract.** This study introduces the Systematic Correlation Matrix Evaluation (SCoMaE) method, a bottom-up approach which combines expert judgment and statistical information to systematically select transparent, non redundant indicators for a comprehensive assessment of the state of the Earth system. The methods consists of two basic steps: 1) Calculation of a correlation matrix among variables relevant for a given research question, and 2) Systematic evaluation of the matrix, to identify clusters of variables with similar behavior and respective mutually independent indicators. Optional further analysis steps include: 3) Interpretation of the identified clusters, enabling a learning effect from the selection of indicators, 4) Testing the robustness of identified clusters with respect to changes in forcing or boundary conditions, 5) Enabling a comparative assessment of varying scenarios by constructing and evaluating a common correlation matrix, or 6) Inclusion of expert judgment such as to prescribe indicators, to allow for considerations other than statistical consistency. The exemplary application of the SCoMaE method to Earth system model output forced by different $CO_2$ emission scenarios reveals the necessity of re-evaluating indicators identified in a historical scenario simulation for an accurate assessment of an intermediate-high, as well as a business-as-usual, climate change scenario simulation. This arises from changes in prevailing correlations in the Earth system under varying climate forcing. For a comparative assessment of the three climate change scenarios, we construct and evaluate a common correlation matrix, in which we identify robust correlations between variables across the three considered scenarios.

# 1 Introduction

An indicator is a quantitative value, measured or calculated, that describes relevant aspects of the state of a defined system. A useful indicator should fulfill certain characteristics that depend on the purpose of the indicator (Gallopín, 1996). Environmental indicators are developed based on quantitative measurements or statistics of environmental conditions in order to allow for a comparison of states of the environment across time or space (Ebert and Welsch, 2004). For environmental indicators of climate change Radermacher (2005) defined statistical measurability, political and societal relevance and scientific consistency, i.e. a scientifically meaningful link between indicator and the state to be described, as three main characteristics that should be considered as important during the indicator selection process. Moreover, other characteristics can be defined as desirable for such indicators, such as a high signal-to-noise ratio of the measurement, the relevance for ethical considerations, or the fact that the chosen indicators should not provide redundant information.

For the assessment of ongoing climate change, models representing the physical and biogeochemical processes of the Earth system, so called Earth system models (ESMs), are one of the essential tools, because the inertia of the climate system to (carbon) perturbations requires projections of future climate states. Early climate models applied simple zero to two dimensional calculations to assess the effect of atmospheric $CO_2$ on the climate by using global mean surface air temperature (SAT) as an indicator (e.g., Arrhenius, 1896; Callendar, 1938; Sellers, 1969). This commonly used climate change indicator, SAT, fulfills all three above mentioned characteristics: Several long term temperature records as well as proxies for assessing SAT exist, which makes this indicator well measured (Statistical measurability). SAT is closely linked to other climate variables, e.g. evaporation, sea level rise, or biological productivity. Although SAT may not be the most relevant variable for society, using this indicator as a proxy for climate impacts is scientifically consistent (Seneviratne et al., 2016). Its political, economical and ethical relevance evolved over time and is now evident in discussions concerning e.g. global warming (Ott et al., 2004) or the 2-degree temperature increase target, which was endorsed by the Conference of the Parties in 2015 (UNFCCC, 2015). Working group II of the Intergovernmental Panel on Climate Change (IPCC) (Houghton et al., 2001) used SAT as the main climate change indicator, due to its predominance in the existing literature and its large scientific consistency as such.

But as Earth system models and observational data sets continuously increase in complexity there are more and more variables available that could potentially serve as indicators for the state of the climate system. Which ones should we select for a fully comprehensive assessment of changes in the climate system, ideally, without providing redundant information? A common bottom-up approach for measuring complex systems is to start from a broad set of (Earth system) variables and consecutively select more appropriate ones depending on the research question (e.g. Pintér et al. (2005); Kopfmüller et al. (2012)). For science-led climate change assessments reports, such as published by Working Group I of the IPCC, in addition to SAT, nowadays more indicators are selected to evaluate changes in different components of the Earth system, e.g., precipitation, or often precipitation extremes, the Arctic summer sea ice or the rate of ocean acidification etc. As such they are discussed in e.g. the IPCC's summary for policy makers of the recent assessment report of climate change (Stocker et al., 2013).

The selection of a limited number of indicators that support scientific, or political decision making is a major challenge for experts, who in this case have to decide on the relative importance of a variable in relation to others (Rametsteiner et al., 2011).

There exist no unambiguous rules for the selection process Böhringer and Jochem (2007). Any indicator selection or metrics construction from Earth system variables, implies a value and weighting decision, and is applying a weight of zero on any disregarded variable. While the value judgment ideally requires the inclusion of potential end-users or stakeholders, the weighting requires a well-informed and broad participation of scientific disciplines, i.e., expert judgment (Radermacher, 2005). However, selecting one indicator, while disregarding the other is a normative choice (Krellenberg et al., 2010), which can (unknowingly) be biased by e.g. technical knowledge (Rametsteiner et al., 2011). Furthermore, Rametsteiner et al. (2011) point out, that the ad-hoc defined indicators should be subject to reevaluation over time.

In this study we want to introduce a bottom-up indicator selection method, that uses statistical information about variables in addition to expert judgement, thereby attempting to reduced bias in the selection process. Systematic Correlation Matrix Evaluation (SCoMaE) uses information on correlations between variables to identify "clusters" of variables that show similar behavior. We then systematically select scientifically consistent indicators to represent these clusters. The identified indicators are independent and do not provide redundant information. A set of independent indicators hence allows for a more comprehensive science-led assessment of the system under consideration, than a set of correlated indicators. Furthermore, SCoMaE allows for a learning process by providing new information about correlations between the given variables, and hence increases the system understanding.

To illustrate the SCoMaE method, we exemplarily select indicators to answer the following defined research question: 'How are changes in the climate system influenced by the sensitivity of the marine and terrestrial biological system to temperature and $CO_2$?'. This example enables us to 1) illustrate how a correlation matrix can be constructed given a specific research question, 2) identify a comprehensive indicator set, 3) show that an indicator set derived from a one forcing scenario is not necessarily appropriate to a changed forcing scenario, 4) identify a common indicator set valid for multiple forcing scenarios, and finally 5) illustrate how the method could be used in an iterative process including expert judgment or previous knowledge of the given system. These steps will serve as the guideline of this paper.

## 2 Defining the research question for the SCoMaE example case

To illustrate the SCoMaE method, we exemplarily select indicators to answer the research question: 'How are changes in the climate system influenced by the sensitivity of the marine and terrestrial biological system to temperature and $CO_2$?'. While for this question we chose to evaluate perturbed parameter simulations of an intermediate complex Earth system model (see section 2.1 and 2.3 for details), it is possible to apply this method to other data sets to answer different questions. Here are a few exemplary alternative ways to define research questions and use the SCoMaE method to investigate.

1) Our application is comparable with a multi-model ensemble, where each of the perturbed parameters is a slightly different version of the default model. We could hence do the very same analysis as described in Section 3, with e.g. the different models and scenarios simulated in the Coupled Model Intercomparsion Project 5 (CMIP5). The research question 'How are simulated changes in the climate system influenced by multi-model model variability under climate change?' could be answered by this setting.

2) To select indicators to answer the research question 'Which changes in the simulated Earth system are robust throughout state-of-the-art Earth system models?', we again could use the CMIP5 datasets. Here, one would probably want to calculate correlations between time series of different variables. This would give information about similar frequencies of those variables, which in turn hints to similar underlying processes. One could compare Correlation Matrices of one model during different

forcing scenarios, as described in sections 3.3 and 3.4, or check the robustness of the correlations in one time period across models.

3) In that sense SCoMaE could also be applied to calculate correlations of observational time series. Since there is a higher level of noice within this data, it is possible to concentrate the research question on pre-defined time scales and filter the time series of the variables, before applying the SCoMaE method. The indicators would accordingly be selected to answer the underlying

research question 'Which are the independent processes that I need to study for a comprehensive assessment of changes in the climate system of a given frequency band?'.

Coming back to our exemplary application of the SCoMaE method, we now want to shortly explain the model set up and simulations.

## 2.1 Model description

This paper illustrates the SCoMaE method for the example of model simulations performed with version 2.9 of the University of Victoria Earth System Climate Model (UVic ESCM), an Earth system model of intermediate complexity (Eby et al., 2013). It includes schemes for ocean physics based on the Modular Ocean Model Version 2 (MOM2) (Pacanowski, 1995), ocean biogeochemistry (Keller et al., 2012), and a terrestrial component including soil and vegetation dynamics (Meissner et al., 2003). It is coupled to a thermodynamic sea-ice model (Bitz et al., 2001) with elastic visco-plastic rheology (Hunke and

Dukowicz, 1997). The atmosphere is represented by a two dimensional atmospheric energy moisture balance model (Fanning and Weaver, 1996). All model components have a common horizontal resolution of 3.6 ° longitude and 1.8 ° latitude and the oceanic component has a vertical resolution of 19 levels, with vertical thickness varying between 50 m near the surface to 500 m in the deep ocean. Wind velocities used to calculate advection of atmospheric heat and moisture as well as the air-sea-ice fluxes of surface momentum, heat and water fluxes, are prescribed as monthly climatological wind fields from NCAR/NCEP

reanalysis data (Eby et al., 2013). Wind anomalies, which are determined from surface pressure anomalies with respect to pre-industrial surface air temperature, are added to the prescribed wind fields.

A list of the globally aggregated output variables is given in Table A1.

## 2.2 Spin-up and Scenario Forcing

For the default model simulation, the UVic ESCM was spun up with pre-industrial (year 1765) seasonal forcing for over

ten thousand years. All simulations were integrated from 850 until 2005 using historical fossil-fuel emissions and land-use changes, as well as radiative forcing from solar variability and volcanic activity following Eby et al. (2013). Following Keller et al. (2014), continental ice sheets were held constant to facilitate the experimental setting and analyses. Warming from black carbon, indirect ozone effects, and cooling from indirect sulphate aerosol effects were not included. From 2005 onward until

**Table 1.** List of perturbed model input parameters

| Abbreviation | short explanation of parameter perturbation |
|---|---|
| Kv low | lower bound of vertical ocean diffusivity |
| Kv high | higher bound of vertical ocean diffusivity |
| no marine T sens | no marine biological sensitivity to temperature |
| no terr. T sens | no terrestrial vegetation sensitivity to temperature |
| veg q10 low | lower bound of the vegetation Q10 sensitivity |
| veg q10 high | higher bound of the vegetation Q10 sensitivity |
| soil q10 low | lower bound of the soil Q10 sensitivity |
| soil q10 high | higher bound of the soil Q10 sensitivity |
| $CO_2$ fert. zero | no $CO_2$ fertilization effect |
| $CO_2$ fert. low | lower bound of $CO_2$ fertilization effect |
| $CO_2$ fert. high | higher bound of $CO_2$ fertilization effect |
| transp. $CO_2$ sens zero | no $CO_2$ sensitivity of transpiration |
| transp. $CO_2$ sens low | lower bound of $CO_2$ sensitivity of transpiration |
| transp. $CO_2$ sens high | higher bound of $CO_2$ sensitivity of transpiration |
| CN $CO_2$ sens | stoichiometric changes in response to changing ocean carbonate chemistry |

2100 the Representative Concentration Pathway (RCP) 4.5 and 8.5 scenarios from Meinshausen et al. (2011) were implemented as an intermediate and high $CO_2$ emissions driven scenario, respectively.

For the sensitivity analysis performed with the UVic ESCM different model input parameters and parameterizations were perturbed, and for some of them it was necessary to do a new model spin-up to again reach steady-state conditions, apart from 5 that the forcing was the same for all simulations.

### 2.3 Parameter Perturbations

In the following sections the single parameter perturbation experiments, which are used in the example and shown in Figure 1 are explained in detail. We chose these parameters to explore the sensitivity of the UVic ESCM to uncertainties in terrestrial and marine biological productivity with respect to temperature and $CO_2$, since these processes will influence the future carbon 10 cycle. In addition, we perturbed ocean vertical diffusivity, since this is a physical process influencing marine carbon uptake. All parameters were perturbed within physically meaningful ranges, which was evaluated based on their agreement with the time series of the historical global mean air temperature (Figure S5). See Table 1 for a quick overview of the simulations.

**Vertical Ocean Diffusivity**

Small-scale physical mixing (vertical diffusivity or diapycnal mixing) in the ocean is parameterized in all global models 15 because of their resolution. Thus, this important process, which plays a key role in determining ocean circulation and bio-geochemical cycles as well as ocean to atmosphere heat and carbon fluxes, is set by necessity as a single global, or several

regional, values that falls within the range of observational estimates of vertical diffusivity. To test how this affects all model results we varied this parameterization by increasing and decreasing it by 50% (Kv low and Kv high), which is within the range of observational estimates (Duteil and Oschlies, 2011). For these sensitivity analysis the model was spun-up with the corresponding setting for 10000 years, until a new equilibrium climate state was reached.

### Lower Bounds of Biological Temperature Sensitivity

Although biological processes are known to be sensitive to temperature, there is a significant amount of uncertainty in how biology will respond to warming caused by climate change (Friedlingstein et al., 2006; Taucher and Oschlies, 2011). Further-more, there are many different ways to model the effects of temperature on biology and it is unknown which is best for Earth

system model applications. To investigate the lower bounds of the sensitivity of biological processes to direct temperature effects we conduct simulations where direct temperature effects on biology are not included. In order to ensure that global biogeochemical fluxes are as close to present-day ones as possible, flux-weighted global averages for temperature-dependent rates were set for all temperature-dependent functions (see Taucher and Oschlies (2011) for details). This approach was applied separately to marine and terrestrial ecosystems:

a) No marine biological sensitivity to temperature: The results of this analysis can be used to estimate a lower boundary for how marine plankton and how their effect on biogeochemical cycles will respond directly to global warming (no marine T sens). For this sensitivity analysis the model was spun-up with the corresponding setting for 10000 years, until a new equilibrium climate state was reached.

b) No terrestrial vegetation sensitivity to temperature: The results of this analysis can be used to estimate a lower boundary

for how terrestrial vegetation and its effect on the carbon cycle will respond directly to global warming (no terr. T sens). For this sensitivity analysis the model was spun-up with the corresponding setting for 10000 years, until a new equilibrium climate state was reached.

### Vegetation and Soil Sensitivity to Temperature

To further investigate the sensitivity of terrestrial biology to temperature we varied the vegetation and soil Q10 values, which are observationally-derived coefficients that are used to model the biological system rate of change in response to a 10 °C temperature increase. Low and high Q10 values of 1.5 and 3.0 (model default is 2.0), which are within the range of observa-tional estimates (Lloyd and Taylor, 1994), were set to investigate how different terrestrial biological sensitivities to temperature affects the model results (veg q10 low/high and soil q10 low/high). For this sensitivity analysis the model was spun-up with

the corresponding setting for 10000 years, until a new equilibrium climate state was reached.

### $CO_2$ Fertilization of Vegetation

Increasing atmospheric $CO_2$ is thought to stimulate terrestrial carbon uptake, through the process of $CO_2$ fertilization (Matthews, 2007; Keenan et al., 2013). This negative carbon cycle feedback results in reduced atmospheric $CO_2$ concentrations, and has

likely accounted for a substantial portion of the historical terrestrial carbon sink (Friedlingstein et al., 2006). However, the

future strength of $CO_2$ fertilization in response to continued carbon emissions is highly uncertain. In order to test the impact of this uncertainty for future climate change simulations, we followed the approach of Matthews (2007) by scaling the $CO_2$ sensitivity of the terrestrial photosynthesis model. We performed a simulation with no $CO_2$ fertilization effect ($CO_2$ fert. zero), as well as two simulations where we varied the strength of the $CO_2$ fertilization effect by increasing and decreasing it by 50%

($CO_2$ fert. high/low) relative to the default model. No additional model spin-up was needed, since the simulated $CO_2$ fertilization effect only happens when the atmospheric $CO_2$ concentration begins to increase, e.g., from the preindustrial period onward.

**$CO_2$ Sensitivity of Transpiration**

Transpiration by plants is highly sensitive to increases in atmospheric $CO_2$, since plants tend to open their stomata less often

in higher $CO_2$ environments, in order to reduce water loss to the atmosphere. The strength of this effect and its impacts on climate are highly uncertain and has been studied both observation and model based (Keenan et al., 2013; Van Der Sleen et al., 2014; Mengis et al., 2015). To test how strongly this affects simulations of future climate, the amount of transpiration for all plant functional types was scaled after Mengis et al. (2015). In this approach the $CO_2$ fertilization effect is not changed. Three simulations were performed: For the first simulation, transpiration did not change relative to the preindustrial level (transp.

$CO_2$ sens zero). For the other two simulations, the scaled transpiration was increased and decreased by 50% of the amount that the model would simulate with the default setting (transp. $CO_2$ sens high/low) as $CO_2$ changes. No additional model spin-up was needed, since the effect of changing $CO_2$ on transpiration only becomes evident when the atmospheric $CO_2$ concentration begins to increase, e.g., from the preindustrial period onward.

**Stoichiometric Changes in Response to Changing Ocean Carbonate Chemistry**

Mesocosm studies that artificially increase the amount of $CO_2$ in seawater (e.g., climate change experiments) have suggested that the C:N content of marine plankton may be sensitive to changes in carbonate chemistry. The mesocosm study of Riebesell et al. (2007) suggested that as $CO_2$ increases the C:N content of phytoplankton may increases, which is a change that would affect the amount of carbon exported to the deep ocean by biological processes and have an effect on other marine

biogeochemical cycles. To test how this affects all model results we implemented the mesocosm-derived relationship between the atmospheric $CO_2$ concentration and the C:N content of plankton as in Oschlies et al. (2008) (CN $CO_2$ sens). No additional model spin-up was needed, since the effect of changing $CO_2$ on plankton stoichiometry only becomes evident when the atmospheric $CO_2$ concentration begins to increase, e.g., from the preindustrial period onward.

## 3  The Systematic Correlation Matrix Evaluation (SCoMaE) method

### 3.1  Step 1: Calculate the Correlation Matrix

Throughout this study, a variable is defined as a model output or observational time series, whereas we refer to it as an indicator if a variable was selected to represent a certain aspect of the considered system. To obtain a comprehensive, non redundant set of indicators to describe a given system, the first step is to construct a correlation matrix, i.e. a matrix including the correlation

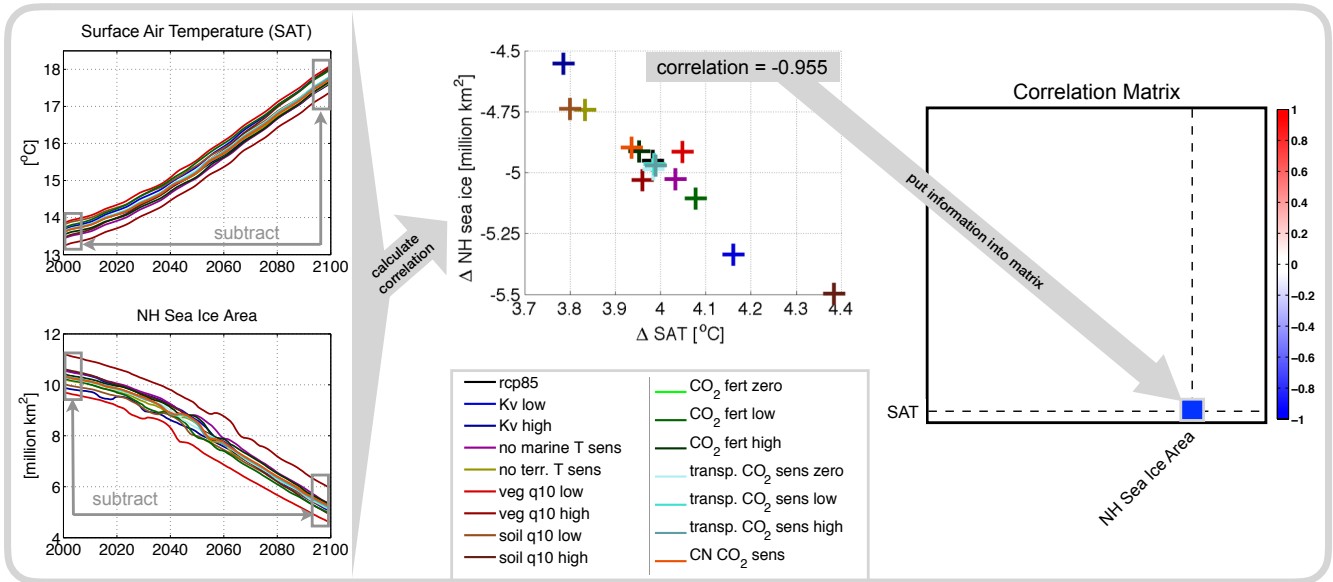

**Figure 1.** Illustration of the Correlation Matrix Construction for the exemplary case study, and the model output variables surface air temperature (SAT) and northern hemisphere (NH) sea ice area. In the first step, temporal differences of the simulations are calculated between 2005-2015 and 2090-2100. Second, changes in the variables induced by the parameter perturbations are correlated. Last, this correlation information is used as one of many entries in the correlation matrix.

information of all the relevant Earth system variables to each other. The construction of the correlation matrix strongly depends on the research question and needs to be adjusted accordingly. The selection of which variables are the relevant variables for the given research question and hence should be included in the matrix, as well as the choice on how the correlations should be calculated is very important for the outcome of the study. In the same way, it is important to consider a reasonable signal to

5  noise ration within the chosen data set. Correlations could for example be calculated between time series of variables or their derivatives, absolute temporal changes, or spatial patterns. Alternatively, output from ensemble simulations could be used to calculate correlations between changes variables due to the different ensemble members. The matrix is then evaluated based on the significance information of these correlations (see Step 2). Note, that for this preselection of the possibly relevant variables to answer the given question, as well as for the construction of the correlation information in the matrix, a certain level of

10  expert judgement is needed.

To illustrate the construction of the matrix based on our example simulations, we show how the correlation between changes in global mean 'surface air temperature' (A_sat) and 'northern hemisphere sea ice area' (O_iceareaN) in the Representative Concentration Pathway (RCP) 8.5 emission scenario (Meinshausen et al., 2011) due to the parameter perturbations translates

15  to the corresponding correlation matrix entry (Figure 1). In our example we want to study the correlations between changes of model output variables, induced by varying poorly constrained model input parameters concerning the carbon cycle. In the

following we will refer to these as 'correlation of variable changes'.

Assuming that the signal of interest is of a similar kind as the state differences between the start and the end of a climate change simulation, we start by calculating the temporal differences between 2005–2015 and 2090–2100 from a number of parameter perturbation simulations that serve as our ensemble in this example (see Section 2.3 for explanations of the parameter perturbations). This enables us to learn whether the different output variables show a similar behavior for the respective parameter perturbation. Then the Pearson correlation coefficients between these changes are calculated and tested by performing a two sided test on a 5% significance level, with $N = 16$, the number of perturbed parameter simulations, and accordingly $t_{crit} = 2.145$.

In our example, there is a negative correlation of variable changes evident between 'surface air temperature' (A_sat) and 'northern hemisphere sea ice area' (O_iceareaN). This illustrates that these model output variables show consistent opposite reactions towards the parameter perturbations, i.e. if the perturbation causes surface air temperatures to increase, it also causes northern hemispheric sea ice to decrease. This information is then written into the correlation matrix. By studying the constructed correlation matrix, and studying single correlations of changes between model output variables, we can learn about basic processes within the simulated climate system, and test if these agree with our expectations. To simplify the visual analysis of our example we sorted the variables in the matrices according to their strength in correlation of variable changes relative to changes in the commonly used climate change indicator, surface air temperature (A_sat) in the historical scenario.

## 3.2 Step 2: Cluster identification and indicator selection

To obtain a set of indicators for the assessment of changes in the regarded system, we systematically evaluate the previously constructed correlation matrix (see Figure 2 for an illustration of this procedure). To obtain a comprehensive, non-redundant indicator set, we follow these steps: 1) The first indicator is the variable with the highest number of significant correlations towards other variables. 2) All variables with a significant correlation are clustered under this indicator. 3) These clustered variables are then excluded from the selection of the next indicator. 4) The next indicators is again the variable with the highest number of significant correlations towards all the remaining variables. 5) This indicator selection procedure is repeated until all variables are clustered and are represented by an indicator. If a variable is not significantly correlated to any of the remaining variables, this variable is considered to be a single indicator. These single indicators are needed for a fully comprehensive assessment, since they show a different behavior from all previously selected indicators and hence provide additional information.

In our example, we applied the SCoMaE method to the correlation matrix concerning 46 commonly used variables for the assessment of climatic changes in the historical forcing scenario, simulated by the UVic ESCM (See Section 2.1 for details on the simulations). We find that the first indicator for our research question in the historical period is precipitation over ocean areas (F_precipO) (Figure 3 and S5). By following the respective column of F_precipO (17th from the right) in the correlation matrix we can see that changes in this model output variable are significantly correlated to changes in all variables that are also significantly correlated to changes in 'surface air temperature' A_sat (1st from the bottom), with the exception of 'mean ocean temperature' (O_temp, 16th from the bottom), but in addition also links changes in global and terrestrial precipitation

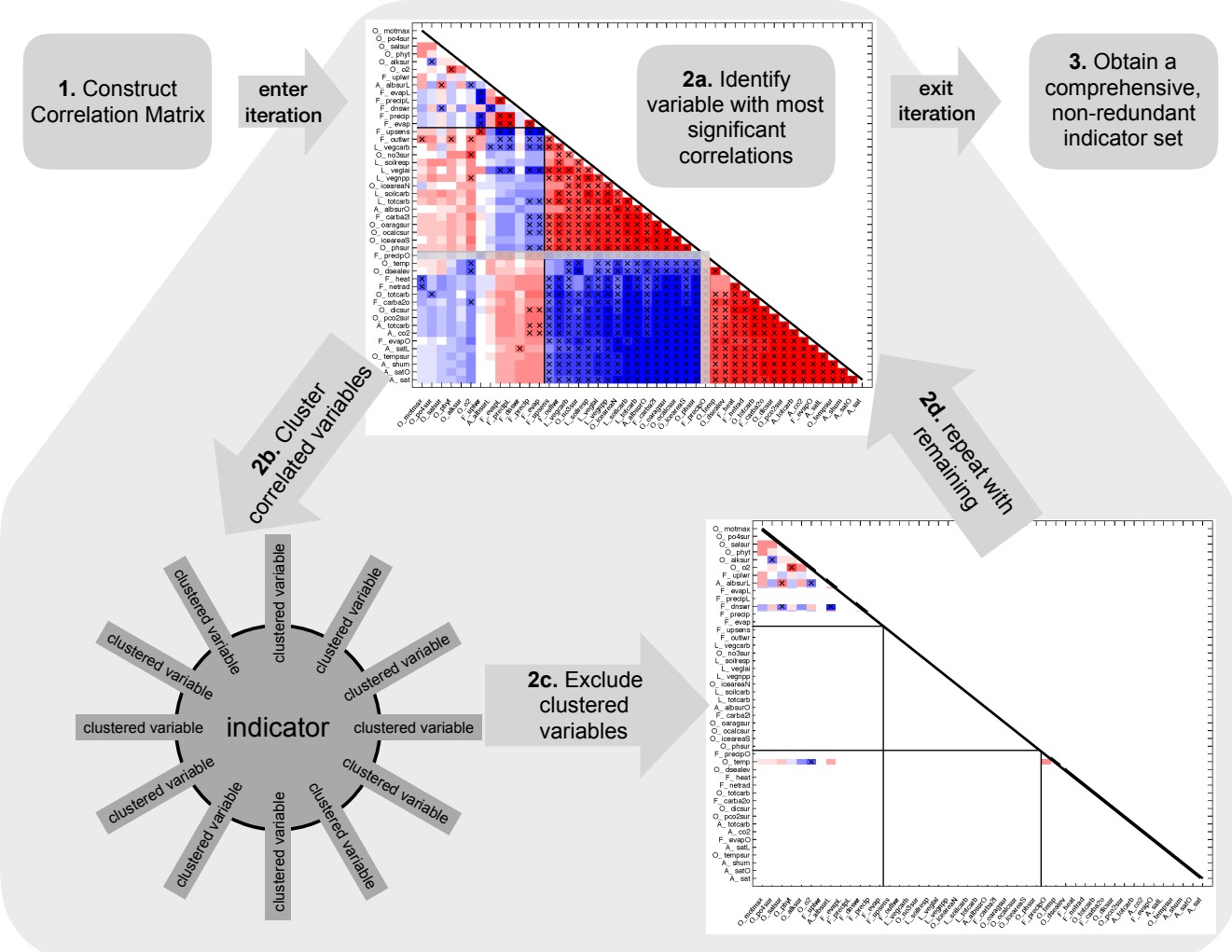

**Figure 2.** Illustration of the indicator selection process, on the example of the correlation matrix for the historical scenario (see Figure 3 for a more detailed display of the correlation matrix). The correlation matrix, was constructed as explained in Figure 1, but for the temporal differences between 1850-1860 and 1995-2005. See Section 3.2 for a detailed step-by-step description of the evaluation process. Prefixes A, O, L and F, stand for atmosphere, ocean, land and fluxes, respectively, for detail description of the model output variables see Table A1

and evapotranspiration (F_precip, F_precipL and F_evap, F_evapL, respectively, and 35th and 37th from the bottom) as well as changes in 'surface net upward longwave radiation' (F_uplwr, 40th from the bottom). The changes of these variables due to parameter perturbations are not significantly correlated to changes in 'surface air temperature' (A_sat). Hence based on purely statistical considerations, using 'precipitation over ocean' (F_precipO) as an indicator for the research question in the historical
5   period, would be preferable to global mean 'surface air temperature' (A_sat), the main ad-hoc indicator for historical climate change, since it potentially holds more information.

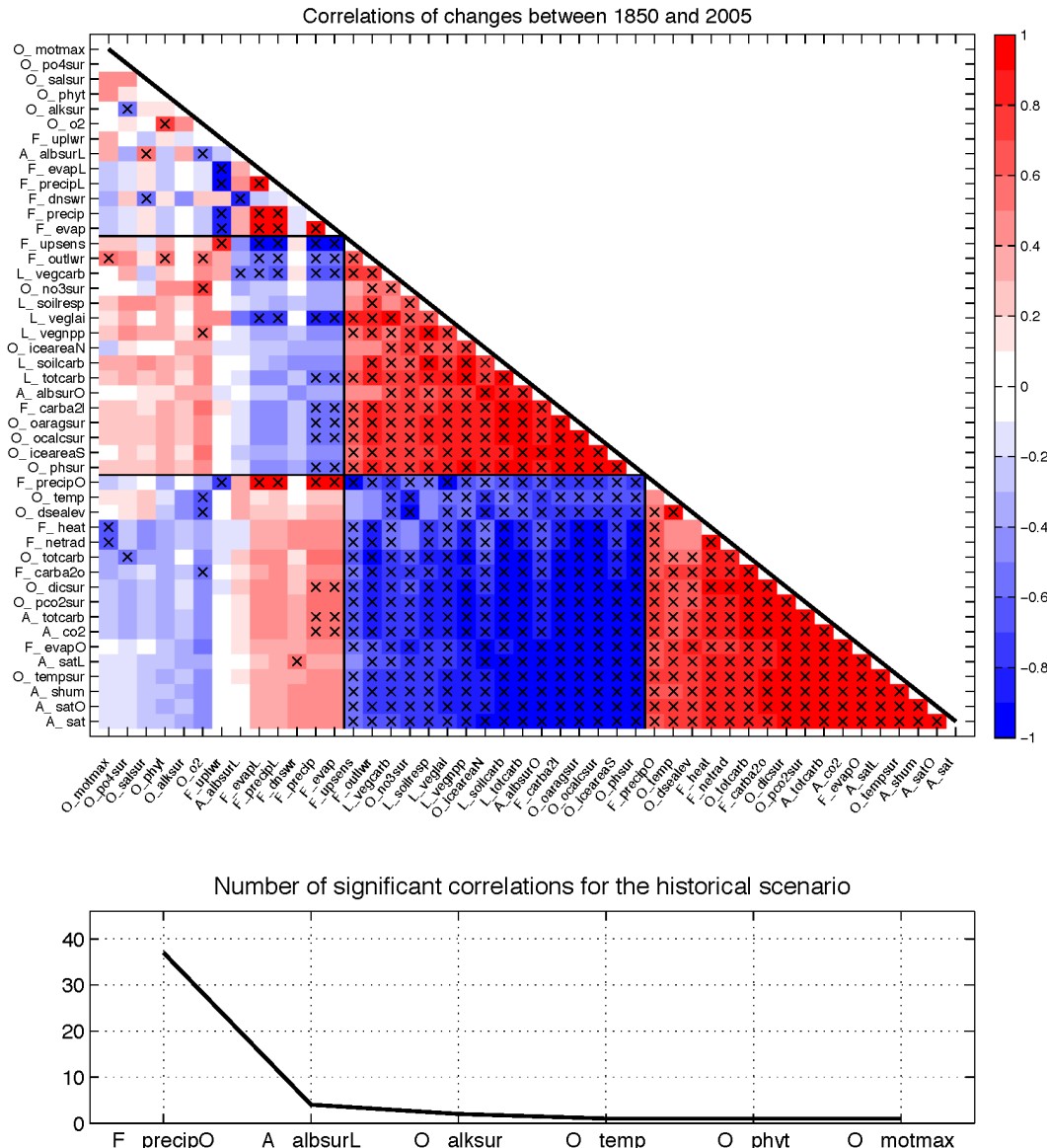

**Figure 3.** top: Correlation matrix for the historical scenario. The correlations are calculated between changes in the 46 model output variables for temporal differences between 1850-1860 and 1995-2005 from the results of the perturbed parameter simulations (as in Figure 1). Correlations significant on a 5% significance level are marked with 'x'. The order of the variables was determined based on their correlation strength to surface air temperature (A_sat) in the historical scenario. Prefixes A, O, L and F, stand for atmosphere, ocean, land and fluxes, respectively, for detail description of the model output variables see Table A1.

bottom: Indicators as identified from the SCoMaE analysis of the correlation matrix above as illustrated in Figure 2, ranked by the amount of significant correlations. The indicators are as follows: precipitation over ocean (F_precipO), land surface albedo (A_albsurL), ocean surface alkalinity (O_alksur), mean ocean temperature (O_temp), ocean phytoplankton (O_phyt) and ocean overturning (O_motmax).

'Surface albedo on land' (A_albsurL) is identified as the second indicator. Its changes due to the parameter perturbations are, after excluding all variables correlated to changes in 'precipitation over ocean' (F_precipO), significantly correlated to changes in 'net surface downward shortwave radiation' (F_dnswr), 'ocean oxygen' (O_o2) and 'sea surface salinity' (O_salsur). The third indicator is 'ocean surface alkalinity' (O_alksur), which shows the same response to the parameter perturbations as 'ocean surface phosphate concentrations' (O_po4sur). When excluding all variables that are clustered under the one of the three above mentioned indicators, three variables remain unclustered: 'Mean ocean temperature' (O_temp), 'maximum meridional overturning' (O_motmax), and 'ocean phytoplankton' (O_phyt). These variables are hence single indicators, which are needed for a comprehensive assessment of the system under consideration (Figure 3b).

See Section 1 and Figures S1 and S2 in the supplementary material, for the results of these analyses for the intermediate-high (RCP4.5) and the business as usual (RCP8.5) scenarios, respectively.

### 3.3 Step 3 (optional): Comparison of indicators for the different forcing scenarios

In order to learn how well the previously identified indicators for one scenario explain a different scenario with changed forcing, we prescribe the use of the previously identified indicator set. The SCoMaE accordingly first uses these indicators and then analyses if and which additional indicators are needed for a fully comprehensive assessment of the new scenario.

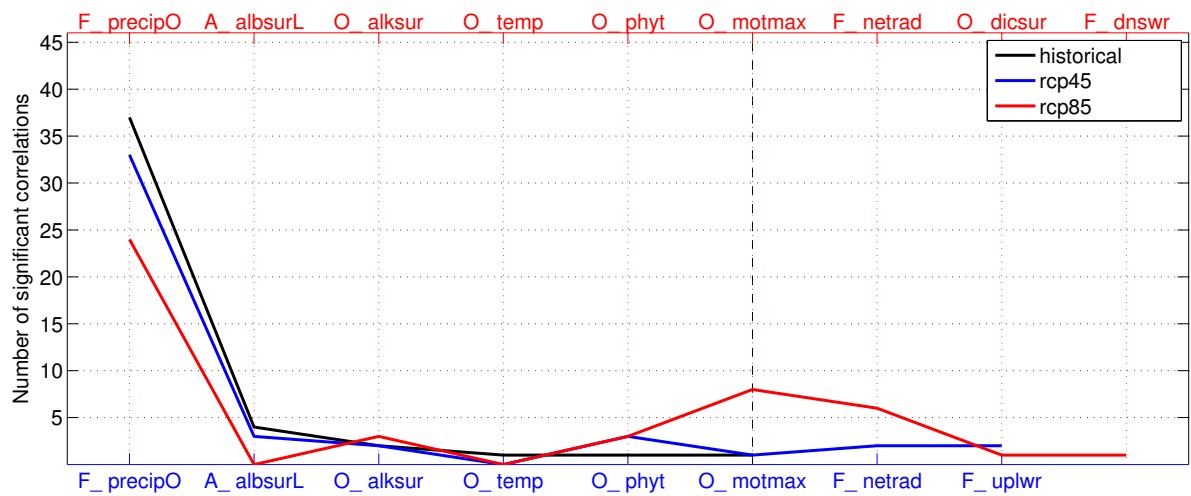

**Figure 4.** Indicators identified from the analysis of the RCP4.5 (blue) and RCP8.5 (red) correlation matrices with the precondition to use the historical indicators first. The indicators are as follows: precipitation over ocean (F_precipO), land surface albedo (A_albsurL), ocean surface alkalinity (O_alksur), mean ocean temperature (O_temp), ocean phytoplankton (O_phyt), ocean overturning (O_motmax), net radiation at the top of the atmosphere (F_netrad), ocean surface dissolved inorganic carbon (O_dicsur) and downward shortwave radiation (F_dnswr).

For the example we prescribed the indicators identified for the historical scenario to assess the intermediate-high (RCP4.5) and the business as usual (RCP8.5) emission scenarios (Figure 4). The results show, that if we were to only utilize the indicators from the historical scenario for the assessment of the two RCP scenarios, we would not be able to assess all changes in the climate system as represented by our model: For the RCP4.5 scenario, we would obtain additional information by considering the variables 'net top of atmosphere radiation' (F_netrad) and 'ocean surface heat flux' (F_heat), which are clustered together, and 'net upward longwave radiation' (F_uplwr) and 'ocean surface salinity' (O_salsur), which form another indicator cluster (Figure 4).

Note, that Earth system variables clustered under the prescribed indicators differ among the different scenarios (compare Figure 3 and S1): In the historical scenario the indicator 'precipitation over ocean' (F_precipO) includes the output variables 'net top of atmosphere radiation' (F_netrad), 'ocean surface heat flux' (F_heat), 'ocean surface nitrate' (O_no3sur), 'top of atmosphere outgoing longwave radiation' (F_outlwr), and 'net upward longwave radiation' (F_uplwr), all of which are not included in the 'precipitation over ocean' (F_precipO) indicator for the RCP4.5 scenario. Instead, the indicator 'precipitation over ocean' (F_precipO) for the RCP4.5 scenario includes 'mean ocean temperature' (O_temp), which it is not included for the historical scenario.

The differences between the correlation matrices for the RCP8.5 scenario compared to the historical scenario are even larger (compare Figure 3 and S2). For the RCP 8.5 scenario eight out of 46 considered variables would not be included if we applied the indicators identified for the historical scenario. Instead we need three additional indicators for the assessment of the system under consideration, namely 'net top of atmosphere radiation' (F_netrad), 'ocean surface dissolved inorganic carbon' (O_dicsur), and 'net surface downward shortwave radiation' (F_dnswr) (Figure 4). Note that six of the eight remaining variables that were initially included in the first indicator cluster for the historical scenario, namely 'precipitation over ocean' (F_precipO), are no longer significantly correlated to it for the RCP8.5 scenario.

These differences in the correlation matrices for the different forcing scenarios indicate changes in prevailing correlations between Earth system variables with the imposed climate forcing. This illustrates that a reevaluation of the chosen indicators may be needed for a comprehensive assessment of different climate strategies yielding different climate states.

## 3.4 Step 4 (optional): Evaluation of a common correlation matrix

To advance this analysis such that changes in correlation matrices from different forcing scenarios can be taken into account, it is possible to create a correlation matrix representing only those correlations that are significant in all forcing scenarios, a so-called common correlation matrix. Applying the SCoMaE method to such a common correlation matrix identifies an indicator set that can be used to assess and also compare multiple scenarios, and hence differs from the previously identified sets for the individual correlation matrices.

To obtain a common indicator set for the three exemplary forcing scenarios (historical, RCP4.5 and RCP8.5), we construct a correlation matrix in which only correlations of variable changes that are significant in all these scenarios are considered (Figure 5). Furthermore the color shading indicates in which of the scenarios the correlations between variable changes were

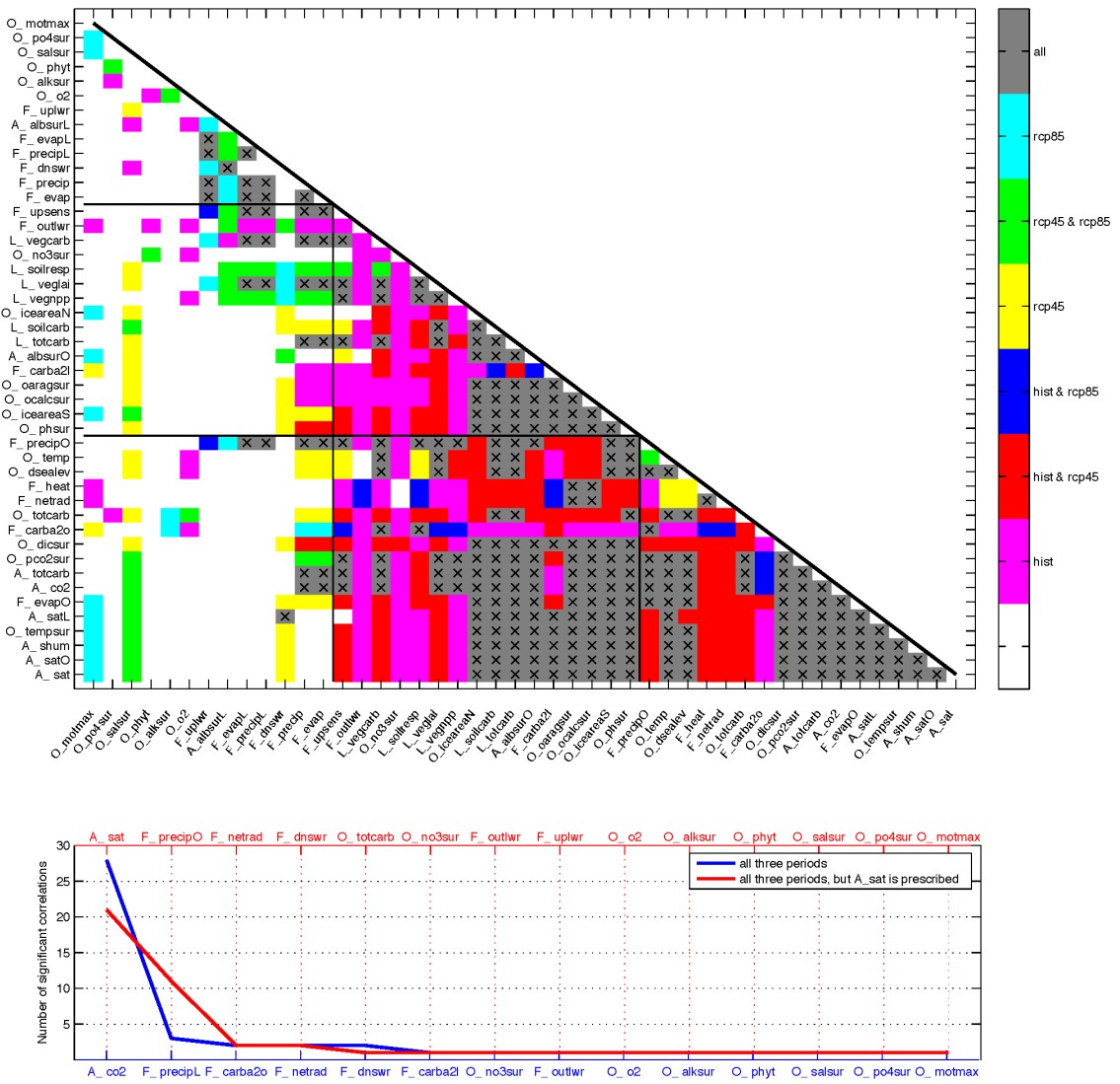

**Figure 5.** a) Correlation matrix for all three scenarios, merging the significance information from all three scenarios. Colors indicate in which scenario the changes of variables due to parameter perturbations showed a significant correlation, see colorbar for explanation. The 'x's mark combinations of variables, where the correlation of variable changes is significant on a 5% significance level in all three scenarios. For details on the regarded model output variables see Table A1.

b) indicators as identified from the analysis based on the correlation matrix above against the number of significant correlations (blue) and with the condition, that surface air temperature (A_sat) is prescribed as the first indicator (red). The indicators are as follows: atmospheric carbon content (A_co2), precipitation over land (F_precipL), atmosphere to ocean carbon flux (F_carba2o), net top of atmosphere radiation ( F_netrad), net surface downward shortwave radiation (F_dnswr), atmosphere to land carbon flux (F_carba2l), ocean surface nitrate (O_no3sur), top of atmosphere outgoing longwave radiation (F_outlwr), ocean oxygen (O_o2), ocean surface alkalinity (O_alksur), ocean phytoplankton (O_phyt), ocean surface salinity (O_salsur), sea surface phosphate (O_po4sur), ocean overturning (O_motmax), precipitation over ocean (F_precipO), ocean carbon (O_totcarb), and surface net upward longwave radiation (F_uplwr).

found to be significant.

A first visual evaluation of the common correlation matrix shows more reddish than bluish shading, which indicates that the correlation patterns for the historical and RCP4.5 scenarios are more similar, than for the historical and RCP8.5 scenarios (Figure 5). This means that for a lower future emission scenario, the indicators from the historical scenario are more suitable than

for a higher future emission scenario. This is true with the exception of the terrestrial and oceanic carbon fluxes (F_carba2l and F_carba2o, respectively). These two fluxes are perturbed by the land-use scheme implemented in the RCP4.5 scenario, since this scenario includes a high amount of afforestation and reforestation. Furthermore, greenish shading shows correlations of variable changes that are significant only in the RCP scenarios, indicating that those correlations of variable changes depend on the increasing anthropogenic (mainly $CO_2$) forcing, included only in these scenarios.

The first indicator obtained from the common SCoMaE analysis is 'atmospheric $CO_2$' (A_co2), which was also found to be the first indicator in the RCP8.5 scenario (Figures S7 and S8). Its changes are significantly correlated to changes in 27 other output variables in all three scenarios. This indicates that these correlations of variable changes are robust throughout the different strength of $CO_2$ forcing in the three scenarios. The fact that 'atmospheric $CO_2$' (A_co2) is the first indicator with a large number of correlated variables, hints at its ability to reflect the changes of other Earth system variables to the param-

eter perturbations, such as changes in temperatures, carbon fluxes and moisture fluxes over the ocean. This can possibly be explained by the fact that the changes in those variables are sensitive to the imposed $CO_2$ forcing, which in turn is reflected in the atmospheric carbon concentration.

The second indicator is 'precipitation over land' (F_precipL), which is clustered with 'terrestrial evapotranspiration' (F_evapL) and 'net upward longwave radiation' (F_uplwr) (Figure S6). This cluster accordingly represents changes in terrestrial moisture

fluxes and the resulting surface upward fluxes of longwave radiation. The latter relates to the surface air temperature, which on land is strongly influenced by the amount of evapotranspiration, and the resulting evaporative cooling. Note, that the fact that terrestrial moisture fluxes are clustered under a separate indicator, hints at a different sensitivity of these variables to the perturbed parameters. Since these three variables show significant correlations of variable changes to each other in all three scenarios, one could use any of them as the indicator for this cluster. The same is true for the next indicators and their clusters,

which are 'air to sea carbon flux' (F_carba2o) and 'soil respiration' (L_soilresp); 'net top of atmosphere radiation' (F_netrad) and the 'ocean surface heat flux' (F_heat); and 'net surface downward shortwave radiation' (F_dnswr) and the 'land surface albedo' (A_albsurL).

The remaining single indicators are 'air to land carbon flux' (F_carba2l), 'ocean surface nitrate' (O_no3sur), 'top of atmosphere outgoing longwave radiation' (F_outlwr), 'ocean oxygen' (O_o2), 'ocean surface alkalinity' (O_alksur), 'ocean phytoplankton'

(O_phyt), 'sea surface salinity' (O_salsur), 'ocean surface phosphate' (O_po4sur) and 'maximum ocean meridional overturning' (O_motmax).

## 3.5  Step 5 (optional): Including expert judgment

If stakeholders or experts were to inform the indicator selection process, it is possible to prescribe indicators and then use the SCoMaE analysis to identify additional uncorrelated variables, that are needed to obtain a comprehensive assessment of the system. Also instead of using global mean time series, one could look at time series of regions or already processed variables, such as heat stress or cumulative emissions. This approach in combination with the SCoMaE analysis enables us to learn about variables, which have previously been disregarded but potentially provide new information about the system, or to learn which of the previously regarded indicators actually provide redundant information.

How would the common indicator set from our example change, if we were to include the condition that 'surface air temperature' should be the first indicator, instead of 'atmospheric $CO_2$'?
Prescribing 'surface air temperature' (A_sat) as the first indicator for the common correlation matrix, leads to the replacement of 'precipitation over land' (F_precipL) by 'precipitation over ocean' (F_precipO) as the second indicator (Figure 5, bottom), its changes to the parameter perturbations is correlated with 12 variables that are clustered under this indicator. Almost all of these variables were initially clustered under 'atmospheric $CO_2$' (A_co2), but are not significantly correlated to changes in 'surface air temperature' (A_sat). These variables mainly describe global and oceanic moisture fluxes, as well as carbon fluxes or reservoirs on land: 'precipitation over the ocean' (F_precipO), 'global evaporation' (F_evap), 'global precipitation' (F_precip), 'vegetation net primary productivity' (L_vegnpp), 'leaf area index' (L_veglai), 'vegetation carbon' (L_vegcarb) and the 'surface upward sensible heat flux' (F_upsens). The only exception to this behavior is 'total ocean carbon' (O_totcarb), which in turn becomes a single indicator. In addition the second indicator, 'precipitation over the ocean' (F_precipO), now incorporates the previously identified clusters of the second and third indicators, namely the clusters of 'precipitation over land' (F_precipL) and the 'air to sea carbon flux' (F_carba2o). Only 'net upward longwave radiation' (F_uplwr), which was also clustered under 'precipitation over land' (F_precipL) becomes a single indicator, remaining unclustered when 'surface air temperature' (A_sat) is prescribed as the primary indicator. In turn 'air to land carbon flux' (F_carba2l), which was a single indicator in the default SCoMaE analysis, is now clustered under 'surface air temperature' (A_sat).
The third and fourth indicators are 'net top of atmosphere radiation' (F_netrad) and 'net surface downward shortwave radiation' (F_dnswr), which were found with the same underlying clusters in the default analysis (compare Figures S8 and S9). Finally, eight of the nine previously identified single indicators, remain unclustered and hence are still single indicators.
Although the total number of indicators has not changed, the identified clusters and their meaning differ: In the default analysis the first indicator represented changes in temperatures, carbon fluxes and global and oceanic moisture fluxes. If 'surface air temperature' (A_sat) is prescribed the global and oceanic moisture fluxes are moved to the second cluster, which in addition incorporates some Earth system variables from the previously identified second and third indicators. This is one example showing how the SCoMaE method allows for the inclusion of expert judgment or preconditions, is able to account for changes in correlation patterns, and allows one to determine which indicators are needed for a comprehensive and non-redundant assessment. (For more discussions see Section 2 and Figure S3 in the supplemental information.)

## 4 Discussion

### 4.1 Discussion of the results from the example

**What could we learn from the example?**

As illustrated above, the SCoMaE method statistically evaluates the correlations between changes of model output variables
and uses this information to cluster variables, while selecting a representative indicator for each cluster. The exemplary anal-
yses of the individual scenarios illustrates the dependence of the indicator selection on the imposed forcing scenario. These
results demonstrate that for our model it is insufficient to apply the historical indicator set to the future scenarios with either
higher $CO_2$ forcing such as in the RCP8.5 scenario, or more limited $CO_2$ forcing and reduced anthropogenic land use such as
in the RCP4.5 scenario. Although our analysis is too limited to conclusively determine a best set of climate change indicators in
a purely scientific bottom-up approach, our results do suggest that a comprehensive assessment of future climatic states needs
a re-evaluation of the ad-hoc chosen indicators, due to changes in prevailing climate responses.

We demonstrate one possible approach for selecting a more comprehensive indicator set by constructing a common correlation
matrix to identify indicators that can be used for the assessment of all three scenarios. For the clusters of variables of the
common indicator set, the correlations of variable changes remain significant even under different atmospheric carbon or land
use forcing.

However, one should always ask if the identified clusters and indicators are scientifically meaningful? For the common cor-
relation matrix (as well as the RCP8.5 scenario), the first indicator, 'atmospheric $CO_2$' (A_co2), groups together variables
describing changes in carbon fluxes, temperatures, and moisture fluxes over the ocean. This is scientifically meaningful, since
changes in carbon fluxes will affect the atmospheric carbon content and hence atmospheric temperatures, both over land and
ocean. These temperature changes in turn have an effect on the moisture fluxes over the ocean, such as the evaporation over
ocean, which is physically driven by temperature changes. These categories are hence physically linked and it is to be expected
that they are correlated irrespective of the chosen forcing scenario.

The second indicator, 'precipitation over land' (F_precipL), represents the variability of moisture fluxes on land and the as-
sociated cooling effect. The fact that these processes are clustered under a indicator that is distinct from global and oceanic
moisture fluxes indicates different underlying processes for these moisture fluxes, namely the influence of biological transpira-
tion. This process is directly affected by the parameter perturbations concerning the sensitivity of transpiration to $CO_2$ (Mengis
et al., 2015) and the $CO_2$ fertilization effect (Matthews, 2007). Given the regarded parameter sensitivities of the model, the
distinction between terrestrial and marine moisture fluxes is scientifically meaningful.

Another identified cluster is 'net top of atmosphere radiation' (F_netrad) and 'ocean surface heat flux' (F_heat), which are
directly linked in the model. Furthermore 'net surface downward shortwave radiation' (F_dnswr) and 'land surface albedo'
(A_albsurL) are clustered, since changes in vegetation on land induced by the parameter perturbations, influence both the sur-
face albedo on land and the incoming shortwave radiation at the surface.

The 'air to sea carbon flux' (F_carba2o) and 'soil respiration' (L_soilresp) are clustered together for all three scenarios, but

show a negative correlation of variable changes in the historical scenario and positive correlations of variable changes in the two RCP scenarios, indicating a dependency on the atmospheric carbon concentrations. The predominant parameterization for those correlations of variable changes is one that affects the $CO_2$ fertilization (Figure S7). Since this is not an intuitive connection, we will shortly discuss this correlation in more detail: The strength of the $CO_2$ fertilization determines the increase of

plant net primary production (NPP) to increasing atmospheric $CO_2$ concentrations. For the historical scenario, in the case when the $CO_2$ fertilization parameterization is increased, soil respiration increases due to an increase in vegetation and hence the soil carbon pool. In the same case the air to sea carbon flux slightly decreases due to lower atmospheric carbon concentration in case of increasing vegetation NPP and consequently land $CO_2$ uptake. Hence the negative correlation of variable changes between the 'air to sea carbon flux' (F_carba2o) and 'soil respiration' (L_soilresp) for the $CO_2$ fertilization perturbation in the

the historical scenario (Figure S7a).

In contrast, in the future, high $CO_2$ and temperature scenarios both Earth system variables show larger changes with increased $CO_2$ fertilization parameterization. For 'soil respiration' (L_soilresp) the underlying process remains the same in this case. However, the terrestrial carbon reservoir reaches a saturation state during the high emission scenarios. With increasing $CO_2$ fertilization strength the land carbon reservoir reaches this saturation state earlier, causing more carbon to remain in the atmo-

sphere, which following Henry's law results in an overall higher 'air to sea carbon flux' (F_carba2o) in the simulations with higher $CO_2$ fertilization, since the ocean equilibrates with the atmosphere. This explains the positive correlation of variable changes under the two RCP scenarios.

Two clusters are identified in both future emission scenarios, namely 'ocean phytoplankton' (O_phyt), which is clustered with 'ocean surface phosphate' (O_po4sur) and 'ocean surface nitrate' (O_no3sur), and 'ocean oxygen' (O_o2), which is clus-

tered with 'ocean surface alkalinity' (O_alksur) (compare Figures S6 and S7). These two clusters are only identified when atmospheric $CO_2$ concentrations are high, but do not hold for the historical scenario, where other relationships seem to be of greater importance. As a result, all of these variables are unclustered for the common indicator selection, causing the number of selected indicators for a common indicator set to increase.

**Limitation of the analyses from the example**

For our case study, we chose to assess the uncertainty of the biological system towards increasing temperature and $CO_2$, which is reflected in the choice of the considered perturbed parameters. In addition to directly perturbing biological parameterizations, we also perturbed some key physical parameters that indirectly influence the biological systems. All parameter perturbations were chosen because the parameterizations are poorly constrained and under future high $CO_2$ and temperature forcing, it will be become increasingly important to take this uncertainty into account. These choices, however, bias the correlation analysis

of the model output variables towards their sensitivity to the selected perturbations (for a detailed discussion see Section 3 and Figure S4 in the supplemental information). For a more comprehensive assessment of uncertainties, experiments with different uncertainties in the simulated Earth system, such as cloud parameterizations or the model's climate sensitivity would need to be considered. Such experiments would accordingly change the patterns of the correlation matrix. Furthermore, to take into account the non-linearity of the Earth system, as a follow-up study, one could co-vary the parameters. This would more realis-

tically reflect the inherent process uncertainty within an Earth system model.

It is important to stress the fact that the Earth system variables used in our example are annual global integrals or means, between two fixed points in time. While our approach was sufficient to demonstrate the SCoMaE method, it is important to mention that global integrals and means are not always positively correlated to regional changes and therefore, may misrep-

resent regional responses. Furthermore, we are not assessing the temporal development of the model variables response to changes in the climate state. But rather investigate changes imposed by parameter perturbations, which are sensitive to $CO_2$ and temperature, under different climate forcing scenarios. This approach was chosen since the UVic ESCM is a model with low internal variability and would hence, likely overestimate information if we were to evaluate temporal correlations. Investigating the model's sensitivity to the parameter perturbations was therefore deemed a better choice for illustrating the SCoMaE

method. Any more thorough climate change assessment using the SCoMaE method would also need to investigate how variable correlations and indicator clusters might change spatially and temporally.

## 4.2  Discussion of the SCoMaE method

The construction of an individual or a common correlation matrix can be a useful tool for assessing the state of complex systems. Individual correlation matrices allow one to obtain an initial overview of relationships between the different system

variables. Whereas a common correlation matrix shows how changes in the state of a system, imposed by e.g. varying forcing scenarios, influences these relationships. The SCoMaE method then allows us to cluster the variables, based on statistical considerations, to obtain a non-redundant indicator set to guide more detailed analysis.

However, in order to be useful one must carefully select, what information to include in the correlation matrix, which in turn strongly depends on the given research question. This can be illustrated by the implicit choices made for our exemplary case

study, where we regarded correlations of variable changes of globally averaged model output variables given various parameter perturbations. The first choice in this case study was to use global aggregates of the model output. However, if the research focus would be set on, e.g., regional phenomena the correlations for the matrix could also be constructed either between regional aggregates, or based on the correlation strength for a given spatial pattern.

The second choice for the case study, was to regard correlations between changes in model output variables based on their

reaction to a parameter perturbation under changing climate forcing. In stead of using model output, it is also possible to further process the data and calculate derivatives of the model output variables, such as heat stress or cumulative time series. On an other note, using a model with higher internal variability, it would also be possible to regard temporal correlations of a chosen time period. This would, in contrast to the purely process-based parameter perturbations that we regarded in the case study, hold information about the time scales and temporal development of the model output variables, which hints at common

underlying processes in the model. Additionally, if the considered time series showed higher internal variability, it might be conceivable to apply a specific temporal filter to the data, before calculating the correlation matrix. This could allow the distinction of important processes on different time scales, from daily and seasonal to inter-annual or decadal.

In the following we want to discuss the contribution of the SCoMaE method to achieve the three characteristics for indicator selection as introduced by Radermacher (2005). Constructing a correlation matrix enables scientists to comprehensively identify correlations in complex systems, such as the Earth system, both simulated and observed. The application of SCoMaE allows one to identify scientifically consistent sets of indicators, which are independent and hence do not provide redundant

information, to be used in a science-led assessment. This method represents a bottom-up, natural science perspective on indicator selection. It thereby tackles one of the three characteristics as discussed by Radermacher (2005), namely that of scientific consistency.

In our example the SCoMaE method is based on model data, and hence does not account for information about the statistical measurability of the identified indicators. This makes it difficult to directly translate a model-based indicator set to a 'real

world' application. This is for example the case for the first indicator in the historical scenario, 'precipitation over ocean' (F_precipO). The lack of long term historical precipitation measurements over the ocean (New et al., 2001) would prevent this indicator from being used in a 'real world' application. It is however, noteworthy that there is value in the knowledge that this variable could hold information about other Earth system variables, and hence it might be worth improving the observational system.

The third characteristic mentioned by Radermacher (2005) is the political relevance of indicators. Since SCoMaE is a bottom-up, science-led approach for indicator selection, it at first disregards political, ethical and economical considerations. However, these considerations as well as measurability constraints can be included in the analysis. By prescribing a certain indicator, e.g. 'surface air temperature' (A_sat) in Section 3.5, SCoMaE allows us to include expert judgment and enables us to identify the remaining indicators needed for a fully comprehensive assessment. An iterative learning process on which indicators are

societally relevant and scientific consistent as proposed by Oschlies et al. (2017); Singh et al. (2015), would hence allow the SCoMaE method to identify scientifically meaningful, measurable, and politically relevant indicators sets.

## 5    Conclusions

In this study we introduced a bottom-up, correlation-based approach to systematically identify indicator sets for the assessment of complex systems. To demonstrate the SCoMaE method, we applied it to correlation matrices constructed with changes in

Earth system variables of an intermediate complex Earth system model, with which we simulated three forcing scenarios. We were able to identify indicators sets for an assessment of the historical as well as for an intermediate high and a business as usual future emission scenario. The comparison of the three correlation matrices yielded the opportunity to assess changes in correlations between changes in Earth system variables introduced by the imposed forcing. These changes in the correlation patterns also motivated a re-evaluation of the selected indicator sets for the different scenarios. We show that it is not sufficient

to apply the indicator set identified for the historical scenario to the intermediate high, nor the business as usual future emission scenario. This result points to the fact, that the classical procedure of ad-hoc indicators, such as surface air temperature, may work well for certain environmental conditions or scenarios, but possibly not so well for others. That is, the subjective choice of indicators may lead to unintended preferences in the interpretation of different scenarios. By combining the three scenarios

into a common correlation matrix, we could identify correlations between changes in Earth system variables that are robust across the three forcing scenarios. Considering these correlations only, enabled us to identify a common indicator set, which was scientifically consistent and would allow us to comparatively assess the three considered scenarios.

This case study is one example out of many possible applications of the correlation matrix and SCoMaE method. The construction of the correlation matrix can be adjusted to the respective research question, which makes the SCoMaE method a generic and flexible tool. An iterative application of the SCoMaE method offers the user the chance to comprehensively assess complex systems such as the Earth system, while including political, ethical and economical considerations, as well as measurability constrains.

*Data availability.* The model data used to generate the figures will be made available at http://thredds.geomar.de.

*Author contributions.* N.M., A.O. and D.P.K. conceived and designed the experiments. D.P.K. and N.M. implemented and performed the experiments. N.M analyzed the data and wrote the manuscript with contributions from D.P.K. and A.O..

*Competing interests.* The authors declare that they have no conflict of interest.

*Acknowledgements.* The authors thank Wilfried Rickels, Martin Quaas and Christian Baatz for their helpful comments, as well as the participants of the Metrics Workshop of the SPP 1689 in March 2015, Hamburg, for their thoughts on metrics and indicators. This work was funded by the DFG Priority Program Climate Engineering: Risks, Challenges, Opportunities? (SPP 1689).

**Table A1.** List of globally aggregated model output variables considered in this study. Part 1: Atmosphere, Fluxes, and Land

| model output name | description | unit |
| --- | --- | --- |
| A_albsurL | land surface albedo | [1] |
| A_albsurO | sea surface albedo | [1] |
| A_co2 | atmospheric $CO_2$ | [ppm] |
| A_sat | air surface temperature | [°C] |
| A_satL | land air surface temperature | [°C] |
| A_satO | ocean air surface temperature | [°C] |
| A_shum | surface specific humidity | [1] |
| A_totcarb | total atmospheric carbon | [Pg C] |
| F_carba2l | air to land carbon flux | [Pg C $yr^{-1}$] |
| F_carba2o | air to sea carbon flux | [Pg C $yr^{-1}$] |
| F_dnswr | net surface downward shortwave radiation | [W $m^{-2}$] |
| F_evap | global evaporation | [kg $H_2O$ $m^{-2}$ $s^{-1}$] |
| F_evapL | evaporation over land | [kg $H_2O$ $m^{-2}$ $s^{-1}$] |
| F_evapO | evaporation over ocean | [kg $H_2O$ $m^{-2}$ $s^{-1}$] |
| F_heat | ocean heat flux | [W $m^{-2}$] |
| F_netrad | net top of atmosphere radiation | [W $m^{-2}$] |
| F_outlwr | top of atmosphere outgoing longwave radiation | [W $m^{-2}$] |
| F_precip | global precipitation | [kg $H_2O$ $m^{-2}$ $s^{-1}$] |
| F_precipL | precipitation over land | [kg $H_2O$ $m^{-2}$ $s^{-1}$] |
| F_precipO | precipitation over ocean | [kg $H_2O$ $m^{-2}$ $s^{-1}$] |
| F_uplwr | surface net upward longwave radiation | [W $m^{-2}$] |
| F_upsens | surface upward sensible heat flux | [W $m^{-2}$] |
| L_soilcarb | soil carbon | [Pg C] |
| L_soilresp | soil respiration | [Pg C $yr^{-1}$] |
| L_totcarb | total land carbon | [Pg C] |
| L_vegcarb | vegetation carbon | [Pg C] |
| L_veglai | leaf area index | [1] |
| L_vegnpp | vegetation net primary productivity | [Pg C $yr^{-1}$] |

**Table A1.** List of globally aggregated model output variables considered in this study.Part 2: Land and Ocean

| model output name | description | unit |
|---|---|---|
| O_alksur | sea surface alkalinity | [mol m$^{-3}$] |
| O_dicsur | sea surface dissolved inorganic carbon | [mol m$^{-3}$] |
| O_dsealev | change in sea level | [m] |
| O_iceareaN | northern hemisphere sea ice area | [m$^2$] |
| O_iceareaS | southern hemisphere sea ice area | [m$^2$] |
| O_motmax | maximum meridional overturning stream function | [m$^3$ s$^{-1}$] |
| O_no3sur | ocean surface nitrate | [mol m$^{-3}$] |
| O_o2 | ocean oxygen | [mol m$^{-3}$] |
| O_oaragsur | sea surface omega aragonite | [1] |
| O_ocalcsur | sea surface omega calcite | [1] |
| O_pco2sur | sea surface partial $CO_2$ pressure | [ppmv] |
| O_phsur | sea surface pH | [1] |
| O_phyt | ocean phytoplankton | [mol N m$^{-3}$] |
| O_po4sur | sea surface phosphate | [mol m$^{-3}$] |
| O_salsur | sea surface salinity | [1] |
| O_temp | mean ocean temperature | [°C] |
| O_tempsur | sea surface temperature | [°C] |
| O_totcarb | total ocean carbon | [Pg C] |

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
