# Peer review of "Systematic Correlation Matrix Evaluation (SCoMaE) - A bottom-up, science-led approach to identify Indicators"

_Earth System Dynamics, 2017_

## Referee Comment (RC1) · Anonymous Referee #1 · 5 Sep 2017

General Comments:

This paper describes a method to identify indicators of environmental change that have statistical measurability, political and societal relevance, and scientific consistency. Indicator identification follows a novel 3-step process, based on the construction of a correlation matrix and significance of the r values. They use an Earth System Model of intermediate complexity run with 3 scenarios (historical, RCP 4.5 and 8.5) and 16 sensitivity perturbations in an example of their methodology.

While this paper was interesting and the concept was of indicators being used to describe natural and forced states is sound, there were some major issues that I feel

need to be addressed in the introduction and methods sections.

Specific Comments:

First, what is the common way to identify/construct indicators? Why would this methodology be new, novel, different, better, etc? Their argument is one-sided and more background into other methods is needed. It is implied that expert judgment is the only criteria for indicator selection.

This type of methodology was not compared across models, hence results are strongly model specific. Also, the methodology could only be applied if the ability to examine model sensitivity exists through perturbation runs. This would limit applying this method to specific models or would be far too computationally expensive to pursue.

The rationale for using 11-year averages for calculating delta was not given. I can assume that because these are global averages and the model has low internal variability (referenced from page 14 line 20) that 11 years is sufficient? 10-30 year periods are more common. Also, for the calculation of the historical period deltas, what years were used? The title on Figure 3 would suggest that some linear method was used to calculate delta? I could not find specific years used.

Additionally, 5% significance level is extremely low. What is the rationale for using such a low significance value? Why not use an absolute threshold for r instead? Or inflate the ensemble used for the correlation calculation by using more than 1 realization from each perturbation? Indicators from the historical period are interesting in that there are likely very few variables that have a clear signal or significant trend. Is this why the very low 5% significance level was used?

Cluster analysis: Why would this method be preferred over more standard methods of PCA/EOF analysis or step-wise regression to group variables? Can the results be compared? The emphasis seems to be on non-redundancy of variables, but what is the value of non-redundant groupings?

In the discussion section (Page 12, line 27-28), it is stated that "our results do suggest that a comprehensive assessment of future climatic states needs a re-evaluation of the ad-hoc chosen indicators, due to changes in prevailing climatic responses." This is an interesting statement in that it implies that the historical indicators are often used to determine future states. Are there previous studies to compare this to? If there were more details in the introduction, this statement would make more sense and have a stronger impact.

Minor comments:

Page 2, line 23: The citation uses "respectively" but it is not needed.

Page 3, line 2. Is there a reference for this model? There are many references in the appendix section, but there should be one here as well.

The correlation matrix is constructed by taking correlation coefficient of 2 variable deltas under 16 perturbation experiments. I am assuming you are using Pearson correlation coefficient ($r$). This should be stated at least once in the methods.

The caption of Figure 1 could use more detail, and the figure itself could be generalized. If this paper is about the method (and not the model) it should be much cleaner and clearer. For example, if the 16 perturbation runs are not discussed in the methods section, why are they shown in the figure? This critique also applies to Figure 2.

The naming structure of the data was confusing. Prefixes A, O, F, and L were used. I am assuming that A was either atmosphere or absolute, F was flux, and O was ocean? There were also suffixes of O or L, which are ocean and land, and N or S (North and South?). While I understand that these variables were used to showcase the method and as such the naming conventions were not important, it was hard to interpret the results based on variable names.

In the abstract, there are 3 non-optional steps, but in the manuscript Step 3 is listed as "optional".
Page 5 line 25, should there be forward slashes in the variable names?

Figure 5, the title of the x-y correlation 'quilt' has the word "periods". Should this be changed to scenarios instead? It is not clear what is meant here. This isn't a plot of the significance of the changes in periods, but more of the significance of the r values between variable deltas?

Page 15, line 16: what is meant by "natural science-based assessments"? "Natural" is used throughout the manuscript, but I am not sure it is needed.

The discussion in the Supplementary section is very interesting, but I would argue that it doesn't belong in this paper. However, it could and should be used in a follow-up paper.
* * *

---

## Referee Comment (RC2) · Anonymous Referee #2 · 19 Sep 2017

General Comments:

This paper introduces a bottom-up approach (SCoMaE) to select a climate indicator for certain climate related question, and illustrates that to answer the same question, climate indicator under different climate scenarios may be different, and common correlation matrix could be used to assess multi-scenarios question. This topic is suitable for the journal, but clarification and improvement are needed.

The paper used one example to demonstrate SCoMaE. However, the example is not clearly described. What is the scientific question to answer? How are those variables selected? It might be better to include an experiment setting section instead of ap-

pendix, since all Figures are based on the model experiments.

Please clarify the meaning of 'correlation', as shown in Figure 1, the correlation is not simply correlation between two variables, but correlation of two variable correlations under different input parameter scenarios. It is misleading through the whole text when discussing 'correlation of variable A and variable B'. Please modify the whole result and discussion.

Specific Comments:

Page 2, Line 14: a comprehensive assessment of what?

Page 2, last paragraph: Although there are more details on this topic on Page 3, line 12-21, it is not clear here whether the authors mean to use all variables from output of an Earth System Model or only certain selected variables? If variables are selected, how to select the variables regarding to certain scientific question? In addition, how about output of different time frequencies?

Page 3, Line 2: What are those selected indicators presenting in this paper? What is the question to answer here?

Page 3, Line 16-21: This method is bottom-up, but the selectin of variables is still expert judgment, as well as how to process the variable (e.g. monthly average or seasonal average?). And as the author mentioned "The selection...is very important for the outcome of the study". Please comment more on this.

Page 3, line 23-32: Please clarify the example question here, if it is "the correlation between global mean 'surface air temperature' (A_sat) and 'northern hemisphere sea ice area' (O_iceareaN)", then the correlation should be between time series of A_sat and O_iceareaN. If it is "the correlation between model output variables, given their reaction to varying model input parameters", then should compare correlations between time series of A_sat and O_iceareaN under different scenarios (different input parameters).

Page 3, Line 30: should mention Appendix A before Appendix B. Otherwise switch the order of A and B.

Page 4, Figure 1: please don't overlap the labels, as well as in other Figures.

Page 5, line 1: what does the 'negative correlation' indicate to? If it indicates to the locations of all crosses in the top right panel in Figure 1 (positive SAT change associated with negative sea ice change), it is correct. If it indicates '-0.955', the negative correlation of correlations in different input scenarios, it is wrong.

Page 5, Line 5: Figure 3 should show after Figure 2. Or change the order of figures.

Page 5, Line 21: precipitation over ocean areas is the first indicator of what?

Page 5, Line 20-30: Do Figure 2 and Figure 3 also use the same way to calculate correlation as in Figure 1 (r=-0.955)? If so, then those correlations are not correlations between variables, but the correlations of correlations under different input scenarios. If Figure 3 is showing the correlations between variables, I strongly doubt that A_sat (global surface temperature) and F_uplwr (surface upwelling longwave radiation) show no correlation. It is impossible, higher surface temperature results stronger surface upwelling longwave radiation according to black body radiation. If Figure 3 is showing correlations among different input scenarios, it makes sense, as in all input parameter scenarios, black body radiation should be the same. In that way, please change the way of description through the whole text: the color bar is not indicating the correlations of variables.

Page 5, Line 33: why surface albedo on land significantly correlated to ocean oxygen and sea surface salinity?

Figure 2: not clear how many model output variables are tested until Figure 3. Instead of "clustered variables", it might be better to list all variable names. How are those variables selected to answer the question of what is "the correlation between global mean 'surface air temperature' (A_sat) and 'northern hemisphere sea ice area"? Or

other questions? Please clarify.

Why Figure S5-S10 are after reference and tables?

Figure 3: need to explain all the variables.

Page 8, 2.3: please clarify the meaning of correlation first, if the correlation is based on different input parameter scenarios, then the text needs to be modified.

Page 13, Line 27: RCP4.5 and RCP8.5 has more $CO_2$ emission than historical scenarios and higher sea temperature will contain less $CO_2$ gas in the ocean. Therefore according to Henry's Law, larger $CO_2$ gradient over the atmosphere and the ocean will enhance the air to sea carbon flux. In addition, under RCP4.5 and RCP8.5, soil respiration enhanced also due to higher temperature.

Page 14, 3.1.2: The method assumed that two time periods have the same climate sensitivity regarding to the input parameter change. But it is not true. For example, $CO_2$ fertilization effect is different under different temperatures. In addition, how to select the variables for analysis will make a big difference in the result.

---

## Author Comment (AC1) · 16 Nov 2017

General Comments:

This paper describes a method to identify indicators of environmental change that have statistical measurability, political and societal relevance, and scientific consistency. Indicator identification follows a novel 3-step process, based on the construction of a correlation matrix and significance of the r values. They use an Earth System Model of intermediate complexity run with 3 scenarios (historical, RCP 4.5 and 8.5) and 16 sensitivity perturbations in an example of their methodology.

[Figure]

While this paper was interesting and the concept was of indicators being used to describe natural and forced states is sound, there were some major issues that I feel need to be addressed in the introduction and methods sections.

**Thank you for your generally positive evaluation of the applied method. We hope to have addressed all of your suggestions mentioned below to your satisfaction. We feel, that your comments helped us to improve the paper.**

Specific Comments:

First, what is the common way to identify/construct indicators? Why would this methodology be new, novel, different, better, etc? Their argument is one-sided and more background into other methods is needed. It is implied that expert judgment is the only criteria for indicator selection.

**We added two references on indicator selection to give more back ground on this topic and point out why we think this should be improved: "Working group II of the Intergovernmental Panel on Climate Change (IPCC) (Houghton et al., 2001) used [ global mean surface air temperature (SAT)] as the main climate change indicator, due to its predominance in the existing literature and its large scientific consistency as such." (page 2, lines 12-15) "However, selecting one indicator, while disregarding the other is a normative choice Krellenberg et al. (2010), which can (unknowingly) be biased by e.g. technical knowledge (Rametsteiner et al., 2011). In this study we want to introduce a bottom-up indicator selection method, that uses statistical information about variables in addition to expert judgement, attempting to reduced this bias in the selection process." (page 2, lines 29-33)**

This type of methodology was not compared across models, hence results are strongly model specific. Also, the methodology could only be applied if the ability to examine model sensitivity exists through perturbation runs. This would limit applying this method to specific models or would be far too computationally expensive to pursue.

**Based on your comment and also the comments from the second reviewer, we decided to add a section in which we explain in more detail the set up of the example. Here we explain that comparing our perturbed parameter simulations, is similar to comparing a multi-model ensemble. This would be a computationally doable application. However, also the application of this method to temporal correlations is possible. (See new Section 2 on page 3)**

The rationale for using 11-year averages for calculating delta was not given. I can assume that because these are global averages and the model has low internal variability (referenced from page 14 line 20) that 11 years is sufficient? 10-30 year periods are more common. Also, for the calculation of the historical period deltas, what years were used? The title on Figure 3 would suggest that some linear method was used to calculate delta? I could not find specific years used.

**The fact that the model has low internal variability was the reason for the 11-year averages. We identified the years in Figure 2 and 3 more clearly. Thank you for pointing this out.**

Additionally, 5% significance level is extremely low. What is the rationale for using such a low significance value? Why not use an absolute threshold for r instead? Or inflate the ensemble used for the correlation calculation by using more than 1 realization from each perturbation? Indicators from the historical period are interesting in that there are likely very few variables that have a clear signal or significant trend. Is this why the very low 5

**Using a 95% confidence that a correlation is significant, is a standard value often used in scientific studies, and was in our opinion appropriate for the assessment of all three scenarios, where varying signal strengths are present. Given that we only have 16 simulations, this is a quite strict criteria. However, we wanted to make sure that there is a real signal in the correlation between the variables reaction to our parameter perturbations.**

Cluster analysis: Why would this method be preferred over more standard methods of PCA/EOF analysis or step-wise regression to group variables? Can the results be compared? The emphasis seems to be on non-redundancy of variables, but what is the value of non-redundant groupings?

**The point we wanted to make here is that we might be able to obtain very similar information from the variables within one cluster, e.g. sea ice and temperature, these are then redundant and there is more value in regarding variables from another cluster, which we would possibly have neglected due to our possibly unknown bias (see page 2, lines 29-31). The value of non-redundancy is also within not un-proportionally stressing one result while reading multiple correlated indicators. We agree with the reviewer that our method should be compared to other ones. However, we feel that such a comparison is too much for this publication and we are currently working on a follow-up study in which we compare our results to a PCA, to see how similar the results obtained by those methods are.**

In the discussion section (Page 12, line 27-28), it is stated that "our results do suggest that a comprehensive assessment of future climatic states needs a re-evaluation of the ad-hoc chosen indicators, due to changes in prevailing climatic responses." This is an interesting statement in that it implies that the historical indicators are often used to determine future states. Are there previous studies to compare this to? If there were more details in the introduction, this statement would make more sense and have a stronger impact.

**The study by Rametsteiner et al., (2011) points a little bit into the direction, that a reassessment of sustainability indicators with time is important, in case new knowledge is obtained about the Earth system, but in their case also in case of changes in societal preferences. We point to their study in the introduction now (see page 2, line 30), but we think, that the finding of changed correlations in future high CO2 and temperature scenarios and the according implications for**

**indicator selection, is one of the novel points in our study.**

Minor comments: Page 2, line 23: The citation uses "respectively" but it is not needed. **Removed.**

Page 3, line 2. Is there a reference for this model? There are many references in the appendix section, but there should be one here as well.

**We added a reference.**

The correlation matrix is constructed by taking correlation coefficient of 2 variable deltas under 16 perturbation experiments. I am assuming you are using Pearson correlation coefficient (r). This should be stated at least once in the methods.

**We added a cleared explanation of the calculation of the correlations.**

The caption of Figure 1 could use more detail, and the figure itself could be generalized. If this paper is about the method (and not the model) it should be much cleaner and clearer. For example, if the 16 perturbation runs are not discussed in the methods section, why are they shown in the figure? This critique also applies to Figure 2.

**We hope to have addressed these issues by adding a new section 2, which explains the experimental set up as well as the single parameter perturbation, before these figures appear.**

The naming structure of the data was confusing. Prefixes A, O, F, and L were used. I am assuming that A was either atmosphere or absolute, F was flux, and O was ocean? There were also suffixes of O or L, which are ocean and land, and N or S (North and South?). While I understand that these variables were used to showcase the method and as such the naming conventions were not important, it was hard to interpret the results based on variable names.

**We added an explanation of the prefixes at least in Figure 2 and 3. Hoping that this improves the basic understanding, otherwise we made sure to have the ex-**

**planation of the variables names, when ever they are mentioned in the text. Also we added table S1 to the appendix now.**

In the abstract, there are 3 non-optional steps, but in the manuscript Step 3 is listed as "optional".

**In fact, we already regard step 3 as optional, so we changed the text accordingly.**

Page 5 line 25, should there be forward slashes in the variable names?

**We changed the text to make it more clear that we we referring to two variables here.**

Figure 5, the title of the x-y correlation 'quilt' has the word "periods". Should this be changed to scenarios instead? It is not clear what is meant here. This isn't a plot of the significance of the changes in periods, but more of the significance of the r values between variable deltas?

**We changed the heading to read scenarios now, thank you for pointing this out. Also we edited the figure caption hoping this makes it more clear now.**

Page 15, line 16: what is meant by "natural science-based assessments"? "Natural" is used throughout the manuscript, but I am not sure it is needed.

**This was referring to a natural science versus social science assessment. We changed the phasing now to science-led assessment, following the phrasing used in Ramentsteiner et al., 2011.**

The discussion in the Supplementary section is very interesting, but I would argue that it doesn't belong in this paper. However, it could and should be used in a follow-up paper.

**Thank you, we removed everything that is not directly referred to in the main article and may publish some of it later.**

**References: Houghton, J., Ding, Y., Griggs, D., Noguer, M., van der Linden, P., Dai, X., Maskell, K., and Johnson, C.: IPCC Third Assessment Report: Climate Change, 2001 (TAR), Working group II: Impacts, Adaptation and Vulnerability, Chapter 19, 2001. Krellenberg, K., Kopfmüller, J., and Barton, J. R.: How sustainable is Santiago de Chile? Current performance, future trends, potential measures. Synthesis report of the risk habitat megacity research initiative (2007-2011), Tech. rep., UFZ-Bericht, Helmholtz-Zentrum für Umweltforschung, 2010. Rametsteiner, E., Pülzl, H., Alkan-Olsson, J., and Frederiksen, P.: Sustainability indicator development—Science or political negotiation?, Ecological indicators, 11, 61–70, 2011.**

Please also note the supplement to this comment:
https://www.earth-syst-dynam-discuss.net/esd-2017-72/esd-2017-72-AC1-supplement.pdf

**Supplement:**

[revised manuscript text omitted]

---

## Author Comment (AC2) · 16 Nov 2017

General Comments:

This paper introduces a bottom-up approach (SCoMaE) to select a climate indicator for certain climate related question, and illustrates that to answer the same question, climate indicator under different climate scenarios may be different, and common correlation matrix could be used to assess multi-scenarios question. This topic is suitable for the journal, but clarification and improvement are needed.

The paper used one example to demonstrate SCoMaE. However, the example is not

clearly described. What is the scientific question to answer? How are those variables selected? It might be better to include an experiment setting section instead of appendix, since all Figures are based on the model experiments.

Please clarify the meaning of 'correlation', as shown in Figure 1, the correlation is not simply correlation between two variables, but correlation of two variable correlations under different input parameter scenarios. It is misleading through the whole text when discussing 'correlation of variable A and variable B'. Please modify the whole result and discussion.

**Thank you for your constructive comments. We have added a section on the experimental design of our example and moved some of the appendix to the methods section. We hope this clarifies what exactly we are aiming in the example. Also we have added a definition of the term correlation to this section. What we are referring to is the correlation of changes in different variables induced by the parameter perturbations. Note, that if a parameter is causing higher global mean temperatures, the physical response in the model in turn is to reduce the sea ice extent. So there is a physical meaning behind these correlation between changes in variables. We have changed the text, wherever we talk about the example to be more explicit on which correlations we are referring to.**

Specific Comments:

Page 2, Line 14: a comprehensive assessment of what?

**As now more clearly stated in the new Section 2 (page 3) later on, we are here referring to an assessment of the sensitivity of changes in the climate system. We added that also in the text at this position, although here it is still a general point not referring to the example case. "Which ones should we select for a fully comprehensive assessment of changes in the climate system, ideally, without providing redundant information?" (page 2, lines 16-17)**

Page 2, last paragraph: Although there are more details on this topic on Page 3, line 12-21, it is not clear here whether the authors mean to use all variables from output of an Earth System Model or only certain selected variables? If variables are selected, how to select the variables regarding to certain scientific question? In addition, how about output of different time frequencies? Page 3, Line 2: What are those selected indicators presenting in this paper? What is the question to answer here?

**As discusses in Section 3.1.2 "Limitations of the analysis from the example", we are here not considering any temporal correlations apart from total changes between two points in time. This is clearly a limitation, but considering the low internal variability in the model, temporal correlations would have been unproportionally overestimated. Looking at different time frequencies would, for other experiments, be a very interesting question, we elaborate on this now in the new Section 2 (page 3). We explain the variable selection as follows: "In our example we applied the SCoMaE method to the correlation matrix concerning 46 commonly used variables for the assessment of climatic changes in the historical forcing scenario as simulated by the UVic ESCM". (page 9, lines 16-17)**

Page 3, Line 16-21: This method is bottom-up, but the selection of variables is still expert judgment, as well as how to process the variable (e.g. monthly average or seasonal average?). And as the author mentioned "The selection...is very important for the outcome of the study". Please comment more on this.

**Thank you for pointing this out again. Yes, the preselection as well as the processing of the variables still requires expert knowledge about the possibly relevant processes and time scales that might be important to answer the research question. To strengthen this point we added a sentence. We hope that we have also made this point clearer by adding section 2. "Note, that for this preselection of the possibly relevant variables to answer the given question, as well as for the construction of the correlation information in the matrix, a certain level of expert judgement is needed." (page 7, lines 27-28)**

Page 3, line 23-32: Please clarify the example question here, if it is "the correlation between global mean 'surface air temperature' (A_sat) and 'northern hemisphere sea ice area' (O_iceareaN)", then the correlation should be between time series of A_sat and O_iceareaN. If it is "the correlation between model output variables, given their reaction to varying model input parameters", then should compare correlations between time series of A_sat and O_iceareaN under different scenarios (different input parameters).

**Sorry for the confusion, we hope to have addressed these issues in the new section 2, that we added following your suggestions.**

Page 3, Line 30: should mention Appendix A before Appendix B. Otherwise switch the order of A and B.

**In the original manuscript, Appendix A was called at page 3 line 4, and appendix B at the same page line 30. However, this changed, since we moved the appendix to the section 2. We have made sure that during the revision process the Appendices appear chronologically.**

Page 4, Figure 1: please don't overlap the labels, as well as in other Figures.

**Sorry for the subpar figure. We have changed the figure accordingly.**

Page 5, line 1: what does the 'negative correlation' indicate to? If it indicates to the locations of all crosses in the top right panel in Figure 1 (positive SAT change associated with negative sea ice change), it is correct. If it indicates '-0.955', the negative correlation of correlations in different input scenarios, it is wrong.

**The location of the crosses in Figure 1, do give us a -0.955 correlation between the positive SAT change associated with negative sea ice change. So the negative correlation we mention here does refer to the correlation between changes in the two variables. We attempted to make this more clear in section 2.**

Page 5, Line 5: Figure 3 should show after Figure 2. Or change the order of figures.

**We removed this figure reference. The order of the figures are following the logical order of the method and should now be correct.**

Page 5, Line 21: precipitation over ocean areas is the first indicator of what?

**The new section explaining the set up for the example hopefully clarifies this, but we also added text here. "We find that the first indicator for our research question in the historical period is precipitation over ocean areas (F_precipO)" (page 10, lines 10-11).**

Page 5, Line 20-30: Do Figure 2 and Figure 3 also use the same way to calculate correlation as in Figure 1 (r=-0.955)? If so, then those correlations are not correlations between variables, but the correlations of correlations under different input scenarios. If Figure 3 is showing the correlations between variables, I strongly doubt that A_sat (global surface temperature) and F_uplwr (surface upwelling longwave radiation) show no correlation. It is impossible, higher surface temperature results stronger surface upwelling longwave radiation according to black body radiation. If Figure 3 is showing correlations among different input scenarios, it makes sense, as in all input parameter scenarios, black body radiation should be the same. In that way, please change the way of description through the whole text: the colorbar is not indicating the correlations of variables.

**Sorry for the confusion, we have improved figure 1, hoping to make more clear, which information is taken to construct the correlation matrix in the example. We have also corrected the phrasing of the correlation information throughout the text. On Page 5, lines 20-23 in the original text we just wanted to point out, that the output variable surface net upwelling long wave radiation in our model is influenced not only by the global mean surface air temperature, but also by processes like evaporative cooling. These two processes are in turn influenced differently by our chosen parameter perturbations than global mean temperature. Therefore we find a significant correlation e.g. between changes in net upward**

**long wave radiation (F_uplwr) and changes in evaporation over land (F_evapL), but not to changes in temperature. A fact that we learned about during these analysis, and which we might have overlooked, if we had assumed a perfect correlation between those two variables. This is exactly the kind of learning process the correlation matrix analysis enables.**

Figure 2: not clear how many model output variables are tested until Figure 3. Instead of "clustered variables", it might be better to list all variable names. How are those variables selected to answer the question of what is "the correlation between global mean 'surface air temperature' (A_sat) and 'northern hemisphere sea ice area"? Or other questions? Please clarify.

**The correlation between the A_sat and O_iceareaN, is supposed to be one example to explain how the correlations in the matrix have been calculated in this exemplary case study. We edited Figures 1 and 2 to make it more clear.**

Why Figure S5-S10 are after reference and tables?

**Thank you for pointing this out. We changed this.**

Figure 3: need to explain all the variables.

**This is done in the text, whenever we actually refer to the variables, apart from that we provided a table in the appendix and point to it in the figure label.** " Page 8, 2.3: please clarify the meaning of correlation first, if the correlation is based on different input parameter scenarios, then the text needs to be modified.

**Yes, we have edited the text, defining the correlation more clearly. See also the new section 2 and the edited Figure 1 for more details on the correlations.**

Page 13, Line 27: RCP4.5 and RCP8.5 has more CO2 emission than historical scenarios and higher sea temperature will contain less CO2 gas in the ocean. Therefore according to Henry's Law, larger CO2 gradient over the atmosphere and the ocean will enhance the air to sea carbon flux. In addition, under RCP4.5 and RCP8.5, soil

respiration enhanced also due to higher temperature.

**Yes, these two physical processes are acting here, we added a statement on this to the text. But the fact that the correlation between changes in the variables is changing sign between the two regarded time periods is linked to the imposed parameter perturbations, as stated in the text.**

Page 14, 3.1.2: The method assumed that two time periods have the same climate sensitivity regarding to the input parameter change. But it is not true. For example, $CO_2$ fertilization effect is different under different temperatures. In addition, how to select the variables for analysis will make a big difference in the result.

**The parameter perturbations are now described in the section 2.3, and not novel, since we applied previously used parameter perturbations that have been studied for this model. The preselection of the variables for our example is also explained now. "In our example, we applied the SCoMaE method to the correlation matrix concerning 46 commonly used variables for the assessment of climatic changes in the historical forcing scenario, simulated by the UVic ESCM (See Section 2.1 for details on the simulations)." (page 10, lines 12-13).**

Please also note the supplement to this comment:
https://www.earth-syst-dynam-discuss.net/esd-2017-72/esd-2017-72-AC2-supplement.pdf
* * *

---

## Author Response (AR2)

**Editor Decision: Publish subject to minor revisions (review by editor)**
(21 Nov 2017) by Ben Kravitz
Comments to the Author:
The authors have done an excellent job of responding to the reviewers. I have a few additional comments that would be worth addressing, mostly in the form of additional discussion in the manuscript:

1) Because the authors were perturbing parameters, they didn't really need information about how variables change in time. That could have been a separate analysis, which the authors do to some degree. I think this separation could be clearer.

The effects of the parameter perturbations are sensitive to changes in atmospheric temperature and CO2 concentration. Instead of looking at transient climate change scenarios, we could have simulated a more idealized scenario with abruptly changed climate forcing, and studied the differences between the resulting climate states. Since we are regarding only absolute changes between end points of our simulations, our analysis is quite similar to such an experimental setting, without containing any detailed information about the temporal development of the variables. We made sure, that the phrasing throughout the manuscript makes it clear that in our example we are referring to analysis of forcing impacts.

"Furthermore, we are not assessing the detailed temporal development of the model variables response to changes in the climate state. Instead, we investigate changes in the final simulated climate state imposed by parameter perturbations, which are sensitive to $CO_{2}$ and temperature, under different climate forcing scenarios."
(page 19, lines 5-8)

"On another note, using a model with higher internal variability, it would also be possible to regard temporal correlations of Earth system variables over a chosen time period. This would, in contrast to the purely process-based parameter perturbations that we regarded in the case study, hold information about the time scales and temporal development of the model output variables, which, in turn, could hint at common underlying processes in the model."
(page 19, lines 27-30)

2) I think there are a few places where the authors overstep. Higher or lower correlation does not mean more or less information, nor can it tell you which

variables are more preferable to use. The actual relationship between the variables is also important.

We are sorry, if we came on too strong at some places. It is definitively important to consider the relationship between variables. We added a sentence to explain, why a e.g. A_co2 has such a high number of signifiant correlations, which is due to the relationship of the variables.
Also we want to make the point, that just because two variables are correlated, this doesn't mean, that they have to be exclusive in an assessment. It is possible that both should be considered, possibly due to societal or political reasons. We changed the text to make these points more clear to the actual potential of the method.

"The fact that 'atmospheric $CO_2$' (A_co2) is the first indicator with a large number of correlated variables, hints at its ability to reflect the changes of other Earth system variables to the parameter perturbations, such as changes in temperatures, carbon fluxes and moisture fluxes over the ocean. This can possibly be explained by the fact that the changes in those variables are sensitive to the imposed $CO_2$ forcing, which in turn is reflected in the atmospheric carbon concentration."
(page 15, lines 13-17)

"An iterative learning process on which indicators are societally relevant and scientifically consistent as proposed by Oschlies et al. (2017); Singh et al. (2015), would hence allow the SCoMaE method to identify scientifically meaningful, measurable, and politically relevant indicators sets."
(page 20, lines 22-25)

3) The authors do not discuss interaction terms, i.e., what happens if multiple parameters are changed at once. We know the climate system contains nonlinearities, so these interaction terms should not all be zero.

A very good point. And definitely a nice follow up study. We added this example to the discussion.

"Furthermore, to take into account the non-linearity of the Earth system, as a follow-up study, one could co-vary the parameters. This would more realistically reflect the inherent process uncertainty within an Earth system model."
(page 18, line 33 to page 19 line 1)

4) If possible, it would be interesting to discuss some "minimum data requirements" for carrying out this method. For example, the authors were

using two time periods that were fairly far apart. What if they were only 20 years apart? Or 30? Would the results be as clear?

We feel this is a complex question to answer. It depends on the signal to noise ratio within the chosen data sets.

In our case the signal in the historical forcing scenario was smaller than the signal in the two future scenarios. This would probably correspond to an shorter time step for the assessment.
But the question of minimum data requirement, depends so strongly on the research question, that we would rather not make a general statement about this.

We added a sentence on the signal in the data in the paragraph on the correlation matrix construction: "In the same way, it is important to consider a reasonable signal-to-noise ratio within the chosen data set."
(page 8, lines 4-5)

[revised manuscript text omitted]